# Antibodies Induced by Smallpox Vaccination after at Least 45 Years Cross-React with and In Vitro Neutralize Mpox Virus: A Role for Polyclonal B Cell Activation?

**DOI:** 10.3390/v16040620

**Published:** 2024-04-17

**Authors:** Sabrina Mariotti, Giulietta Venturi, Maria Vincenza Chiantore, Raffaela Teloni, Riccardo De Santis, Antonello Amendola, Claudia Fortuna, Giulia Marsili, Giorgia Grilli, Maria Stella Lia, Seble Tekle Kiros, Filippo Lagi, Alessandro Bartoloni, Angelo Iacobino, Raffaele Cresta, Marco Lastilla, Roberto Biselli, Paola Di Bonito, Florigio Lista, Roberto Nisini

**Affiliations:** 1Istituto Superiore di Sanità, Viale Regina Elena 299, 00161 Roma, Italy; sabrina.mariotti@iss.it (S.M.); giulietta.venturi@iss.it (G.V.); mariavincenza.chiantore@iss.it (M.V.C.); raffaela.teloni@iss.it (R.T.); antonello.amendola@iss.it (A.A.); claudia.fortuna@iss.it (C.F.); giulia.marsili@iss.it (G.M.); angelo.iacobino@iss.it (A.I.); paola.dibonito@iss.it (P.D.B.); 2Defense Institute for Biomedical Sciences, 00184 Roma, Italy; r.desantis@uniroma1.it (R.D.S.); giorgia.grilli@persociv.difesa.it (G.G.); mariastella.lia@persociv.difesa.it (M.S.L.); florigio.lista@esercito.difesa.it (F.L.); 3University Hospital Careggi, 50134 Firenze, Italy; sebletekle.kiros@unifi.it (S.T.K.); lagif@aou-careggi.toscana.it (F.L.); alessandro.bartoloni@unifi.it (A.B.); 4Aeronautica Militare, Comando Logistico, Servizio Sanitario, 00185 Roma, Italy; dr.raffaele.cresta@gmail.com (R.C.); marco.lastilla@aeronautica.difesa.it (M.L.); bobbiselli@libero.it (R.B.)

**Keywords:** Mpox virus (MPXV), vaccinia virus (VV), Mpox disease (Mpox), smallpox vaccination, immunological memory, bystander help

## Abstract

Aims: To evaluate whether antibodies specific for the vaccinia virus (VV) are still detectable after at least 45 years from immunization. To confirm that VV-specific antibodies are endowed with the capacity to neutralize Mpox virus (MPXV) in vitro. To test a possible role of polyclonal non-specific activation in the maintenance of immunologic memory. Methods: Sera were collected from the following groups: smallpox-vaccinated individuals with or without latent tuberculosis infection (LTBI), unvaccinated donors, and convalescent individuals after MPXV infection. Supernatant of VV- or MPXV-infected Vero cells were inactivated and used as antigens in ELISA or in Western blot (WB) analyses. An MPXV plaque reduction neutralization test (PRNT) was optimized and performed on study samples. VV- and PPD-specific memory T cells were measured by flow cytometry. Results: None of the smallpox unvaccinated donors tested positive in ELISA or WB analysis and their sera were unable to neutralize MPXV in vitro. Sera from all the individuals convalescing from an MPXV infection tested positive for anti-VV or MPXV IgG with high titers and showed MPXV in vitro neutralization capacity. Sera from most of the vaccinated individuals showed IgG anti-VV and anti-MPXV at high titers. WB analyses showed that positive sera from vaccinated or convalescent individuals recognized both VV and MPXV antigens. Higher VV-specific IgG titer and specific T cells were observed in LTBI individuals. Conclusions: ELISA and WB performed using supernatant of VV- or MPXV-infected cells are suitable to identify individuals vaccinated against smallpox at more than 45 years from immunization and individuals convalescing from a recent MPXV infection. ELISA and WB results show a good correlation with PRNT. Data confirm that a smallpox vaccination induces a long-lasting memory in terms of specific IgG and that antibodies raised against VV may neutralize MPXV in vitro. Finally, higher titers of VV-specific antibodies and higher frequency of VV-specific memory T cells in LTBI individuals suggest a role of polyclonal non-specific activation in the maintenance of immunologic memory.

## 1. Introduction

Mpox (previously monkeypox) is an infection caused by the Mpox virus (MPXV), a double-stranded DNA virus belonging to the Orthopoxvirus (OPXV) genus in the family of Poxviridae that are genetically and antigenically very similar [1]. Although the disease presentation resembles smallpox, caused by the closely related variola virus, Mpox has a lower mortality. Symptoms of the infection include fever, myalgia, lymphadenopathies, and skin and mucosal rash [2,3].

Mpox is a zoonotic disease, but its animal reservoir is uncertain and various rodent species from Central and West African tropical rainforests are suspected candidates [4]. Human-to-human transmission of Mpox also occurs upon contact with infected skin, bodily fluids, contaminated items such as bedding, or sexual intercourse [5,6].

There have been more than 94,000 Mpox cases in the 2022–2023 outbreak (https://www.cdc.gov/poxvirus/mpox/response/2022/index.html, accessed on 10 January 2024) in an unprecedented worldwide spread that led the WHO to declare Mpox a global health emergency [7].

A functional immunological memory is established in Mpox convalescent individuals [8] that may confer some degree of protection to re-infections. MPXV has always been confined to Africa with sporadic outbreaks in America and Europe. Therefore, if natural immunization and a certain degree of protection from Mpox may be hypothesized in limited areas of Africa, the extent of pre-existing immunity in the general population of non-African countries is very limited. Studies have shown that, due to the high sequence homology between MPXV and the vaccinia virus (VV), smallpox vaccination provides at least partial protection from MPXV [3,9] by cross-immunity [10,11,12].

However, global smallpox vaccination campaigns ended in the 1970s and, with some exceptions including military and laboratory personnel, nobody has received smallpox boosters in the last few decades. Therefore, younger individuals are not vaccinated and both the extent of the residual immunological smallpox memory response after many years and the efficacy of its potential protective role against MPXV infection are unclear.

In this work, we set up methods for the quantification of specific antibodies to evaluate the humoral immune response to smallpox after vaccination, and MPXV after infection. The methods were used to detect VV-specific antibodies in individuals after at least 45 years from immunization and in those convalescing from an MPXV infection. The culture of VV and MPXV permitted to test the capacity of VV- or MPXV-specific antibodies to neutralize MPXV in vitro, as a possible surrogate of protection. To test a potential role of polyclonal non-specific activation in the maintenance of immunologic memory [13], we compared the VV-specific humoral and cellular immune response in vaccinated subjects with or without a latent tuberculosis infection (LTBI).

## 2. Materials and Methods

### 2.1. Study Population

Blood was collected from donors that gave their informed consent to participate in the study, according to the protocol approved by the Istituto Superiore di Sanità Ethical Committee. All the donors were male, smallpox-vaccinated individuals (age range: 46–64 years), all presumptively vaccinated with the vaccinia virus Lister strain that was used in Italy in the 1960s and early 1970s, which was produced by the Company Sclavo (Siena, Italy). They were divided into the following three groups: individuals with LTBI [Group 1, N = 21], without LTBI [Group 2, N = 20], and unvaccinated donors (age range 32–44 years) without LTBI [Group 3, N = 12]. The unvaccinated donors were younger than the smallpox vaccinated donors because smallpox vaccination was mandatory in Italy up until 1975. All the volunteers were recruited from military personnel of Italian Air Force, and blood samples were collected during check-ups carried out from January to September 2018, before their deployment in specific missions abroad, including international peacekeeping operation activities. Peripheral blood mononuclear cells (PBMC) were isolated by ficoll-hypaque from heparinized blood and frozen in liquid nitrogen, while aliquots of sera were frozen at −70 °C until use. All the donors selected for the study were in good health, without any ongoing pathology. In particular, vaccinated donors did not receive booster(s) of smallpox vaccine, and none of the donors reported history or showed signs or symptoms of MPXV infection. LTBI was confirmed or excluded by a commercially available Interferon Gamma Releasing Assay (QuantiFERON-TB Gold-in tube blood test) used according to the procedures indicated by the manufacturer (QIAGEN, Hilden, Germany). A setting of 0.35 IU/mL for a cutoff value was used, as recommended by the CDC [14]. All participants who were TB gold positive underwent chest radiography to exclude pulmonary TB. Finally, Group 4 (N = 8) included convalescent patients after an MPXV infection. All patients were male (average age: 34.88 ± 7.41, range: 24–43 years) with an increased risk because of MSM, one was HIV positive, another was affected by chronic inflammatory bowel disease, and none were vaccinated for smallpox. All patients were recruited by the Careggi University Hospital, Florence, Italy, where they received treatment after the diagnosis of Mpox with a positive RT-PCR in August 2022, and the blood samples were obtained in October 2022.

### 2.2. Viruses

MPXV (hMpxV/Italy/LOM-AMC-2205251-DS/2022), isolated from a swab by the Defense Institute for Biomedical Sciences, Rome, Italy, was propagated in Vero E6 cells cultured in MEM supplemented with 2% FBS, 0.2 U/mL penicillin, and 0.2 mg/mL streptomycin (PS, Gibco). At 90–100% cytopathic effect in the cell monolayer, the infected cells were harvested by scraping, resuspended and centrifuged for 5 min at 600× *g*. Cell pellets were resuspended and the cell suspensions were lysed through repeated freeze–thaw at −70 °C to +37 °C in order to facilitate the virus release from the cells. The lysates were centrifuged for 5 min at 600× *g*, and after the removing of the sediment the viral stocks were stored at −80 °C. The viral stocks titer was determined by plaque assay in Vero cells. Briefly, tenfold dilutions (200 μL) of the viral stock diluted in cell cultivation medium (MEM + glutamax + 1% aa + 1% PS + 2%FBS) were aliquoted in quadruplicate on the bottom of 24-well tissue culture plates, and overlayed with 190,000 Vero cells/well in 300 μL of cell cultivation medium. Plates were incubated at 37 °C and 5% CO_2_ for 5 days. To calculate plaque forming units (PFU), the cultivation medium was discarded, and the cells were washed with a physiological solution, followed by staining with a crystal violet solution and a manual count of the plaques. Viral culture and all the experiments with MPXV were conducted in the Biological Safety Level 3 (BSL-3) laboratories of the Defense Institute for Biomedical Sciences and of Istituto Superiore di Sanità, Rome, Italy.

A monolayer of Vero E6 cells at 70% confluence was infected with a trypsinized VV stock (vTF7-3 strain), at approximately 1 PFU/cell in DMEM with 2% heat-inactivated FBS and antibiotics (complete medium), in the incubator at 37 °C and 5% CO_2_ controlled atmosphere [15]. After 1 h of infection, the virus was replaced with complete DMEM. After 48–72 h of incubation, and when the cytopathic effect (CPE) was 80–90%, the medium was removed. The cell-associated virus was harvested in 1 mM Tris-HCl pH 9 hypotonic solution (5 mL in 150 cm^2^ flask) and lysed with repeated freeze–thaw cycles in dry ice. Cell debris were removed by centrifugation for 15 min at 3000 rpm at 4 °C in a swing bucket centrifuge, and the virus-containing supernatant was aliquoted and stored at −80 °C.

This virus stock was titered by a rapid plaque assay in Vero 6 cells. Briefly, the monolayer of Vero cells was trypsinized and diluted at 5 × 10^5^ cell/mL. This cell suspension (0.5 mL) was added to the virus samples (0.1 mL each) of the tenfold serial dilutions of the virus stock treated (1:1) with 0.25 mg/mL trypsin. The cell–virus suspensions were plated in 6-well plates, and 0.5 mL of a semisolid overlay media (carboxymethylcellulose 3% in DMEM) was also added in each well. After 3–4 days of incubation, 1 mL of 10% formalin/PBS was added to each well and incubated for 10 min at room temperature (RT). The medium was then aspirated, and 1 mL of crystal violet (0.1% in 20% EtOH) was added. After 10 min at RT, the wells were washed in H_2_O and allowed to dry. Plaques were then counted, and the viral titer was determined accordingly (3.5 × 10^6^ PFU/mL). The VV stock was inactivated by 1h treatment at 60 °C before being used as a coating antigen or as protein-lysed in ELISA or Western blotting.

Protein lysate was obtained by solubilizing VV with 5 mL PBS, containing 8M urea, and with the cell-associated virus contained in the 150 cm^2^ flask showing 80–90% CPE. After several hours of rotating incubation at RT, the lysate became limpid and only a cloud of DNA was visible and was manually removed. This protein lysate was quantified in ELISA using reference sera specific for VV and used in ELISA and Western blotting. Uninfected Vero cell monolayers undergoing the same procedures were used as mock samples. Modified vaccinia virus Ankara (MVA), that was generously provided by Bavarian Nordic GmbH (Dr. José Medina Echeverz and Dr. Henning Lauterbach), was diluted in RPMI, and used to infect antigen-presenting cells without further treatments.

### 2.3. ELISA

VV- or MPXV-specific antibodies were analyzed by enzyme-linked immunosorbent assay (ELISA), as previously reported [16]. Briefly, 96-well polystyrene plates (Greiner Bio-One, Rainbach, Austria) were coated with 50 μL/well of vTF7-3 or hMpxV/Italy protein lysate at a concentration of 5 μg/mL in carbonate bicarbonate buffer and incubated at +4 °C overnight. After three washes with PBS supplemented with 0.005% Tween 20 (TPBS), the plates were blocked with PBS + 2% BSA for 1 h at RT. After three more washes, 50 μL/well of serial dilutions of serum samples, starting from 1:10 with PBS + 2% BSA, were incubated for 3 h at 37 °C. After three further washes, 50 μL/well of secondary mouse anti-human IgG alkaline phosphatase (PA)-conjugated antibody (Southern Biotech, Birmingham, AL, USA) diluted 1:1000 was added for 1 h at 37 °C. Plates were developed by adding 100 µL/well of a substrate, 4-Nitrophenyl phosphate disodium salt hexahydrate, and stopped with 50 µL of 3 N NaOH. Absorbance (405 nm) was measured, and the results were considered positive if the optical density (OD) was three times greater than the negative control.

### 2.4. Neutralization Assay Plaque Reduction Neutralization Test (PRNT) for MPXV

The assay was performed in 24-well tissue culture plates, using the MPXV strain hMpxV/Italy/LOM-AMC-2205251-DS/2022. Both the virus and serum samples were diluted in serum-free cell cultivation medium (MEM + glutamax + 1% aa + 1% PS); equal volumes (300 μL) of MPXV dilution containing 80 PFU per 100 uL, and of each serum dilution, were mixed, and incubated at 37 °C for 2 h. Serum–virus mixtures (200 uL) were then aliquoted in duplicate on the bottom of 24-well tissue culture plates, and overlayed with 190,000 Vero cells/well in 300 μL of cell cultivation medium supplemented with 4% FBS. Plates were incubated at 37 °C and 5% CO_2_ for 5 days, and then stained with 1.5% crystal violet. A titration of MPXV with three dilutions in duplicate was performed in each assay as a control for the test. Neutralizing antibody titers were calculated as the reciprocal of the serum dilution that gave a 50 or 80% reduction in the number of plaques (PRNT50/PRNT80), as compared to the virus control. PRNT80 ≥ 10 and PRNT50 ≥ 10 titers were considered as positive. For statistical analysis, the PRNT titers were transformed to logarithms.

### 2.5. SDS-PAGE and Western Blot Analysis

Supernatants from the Vero cells not infected, or infected with MPXV or VV, were denatured with 5% β-mercaptoethanol in SDS-loading buffer, containing 50 mM Tris-HCl, pH 6.8, 3% SDS, 50% glycerol, 0.5% bromophenol blue. Samples were then heated at 95 °C for 8 min and loaded onto 4–15% gradient mini-PROTEAN TGX precast gels (Bio-Rad, Hercules, CA, USA), which were blotted onto polyvinylidene difluoride membranes (Thermo Fisher Scientific, Waltham MA, USA) using the Trans-Blot Turbo system (Bio-Rad). The membranes were blocked with 3% skim milk in Tris-buffered saline and 0.05% Tween 20 before incubation with the sera of Mpox convalescent individuals, and VV vaccinated or unvaccinated individuals as a control. Immune complexes were detected by Horseradish Peroxidase (HRP)-conjugated goat anti-human IgG (Sigma-Aldrich, St. Louis, MO, USA) using the Crescendo Western HRP chemiluminescent substrate (Millipore, Burlington, MA, USA).

### 2.6. Monocytes Isolation, Dendritic Cells Generation and In Vitro Cell Infection

PBMC were isolated with a Ficoll density gradient and frozen until use. Monocytes were then positively sorted using anti-CD14-labeled magnetic beads (MACS; Miltenyi Biotech, Bergish Gladbach, Germany) starting from the thawed PBMC and resuspended in RPMI 1640-based complete medium. The flow-through CD14 negative (CD14^neg^) fraction of the magnetic bead separation was collected as the negative fraction depleted of the labeled cells and used as CD14^neg^ responder cells. Half of the monocytes were infected with MVA and the other half were used to generate dendritic cells (DC) by culturing the monocytes for 5 days in complete medium containing 50 ng/mL GM-CSF and 1000 U/mL IL-4 (R&D System, Minneapolis MN, USA). In all the experiments, the monocytes and differentiated DC were infected in a tube with MVA at multiplicities of infection (ratios of virus to cells) of 10:1 for 2 h.

### 2.7. Antigen Presentation Assays

The CD14^neg^ responder cells were labeled with 0.75 μM of carboxyfluorescein diacetate succinimidyl ester (CFSE; eBioscience, San Diego, CA, USA), and then co-cultured with non-infected or MVA-infected monocytes (ratio of responder cell–monocytes = 3:1) in a 24-well plate (Corning Inc. Corning NY, USA) in the presence of 1 mL/well of RPMI 1640 complete medium and 5% human serum (RPMI-HS). After 6 days of co-culture, the DC generated from monocytes of the same donor were either infected or not for two hours with MVA (MOI 10:1) in the tube and then plated at 1.5 × 10^5^ cells/well in a 24-well plate with RPMI-HS. Then, the cells which were primed 6 days before with infected monocytes were collected from the wells, washed, and 7.5 × 10^5^ cells/well were restimulated with infected DC. Two days later, the T cell activation was evaluated by FACS analysis of CFSE staining and the intracellular accumulation of TNF-α and IFN-γ.

### 2.8. FACS Analysis

For intracellular cytokine staining, T cells were treated for 2 h with brefeldin-A (Golgi plug; BD Pharmingen) at a concentration of 2 μg/mL, then harvested and washed in phosphate-buffered saline, containing 1% FBS and 0.1% NaN_3_ (staining buffer). They were then stained extracellularly using PerCP-conjugated anti-CD3, PE-conjugated anti-CD4, APC-Cy7-conjugated anti-CD8 monoclonal antibodies or the appropriate isotype controls for background determination, all purchased from BD Pharmingen (San Diego, CA, USA). Then, the cells were fixed and permeabilized using Cytofix/Cytoperm™ (BD Pharmingen), according to the manufacturer’s instructions, and stained with PeCy7-conjugated anti-human IFN-γ and APC-conjugated TNF-α (BD Pharmingen). Stained cells were then analyzed with flow cytometry using the Beckman Coulter Gallios equipped with Kaluza Software (Beckman Coulter, Brea, CA, USA).

### 2.9. Statistical Analysis

For the neutralization and ELISA experiments, technical duplicates were performed. All the statistical analyses were performed using the GraphPad Prism software v9 (GraphPad Software, San Diego, CA, USA). The statistical significance of the difference between groups of data with a normal distribution was determined by the non-parametric Mann–Whitney U test. All tests for statistical significance were two-tailed and *p*-values < 0.05 were considered significant.

## 3. Results

### 3.1. Development of ELISA Test and Neutralization Assay for the Serological Diagnosis of Mpox

To develop an ELISA for the serological diagnosis of Mpox infection, we prepared and inactivated an MPXV culture supernatant to be used as the ELISA plates’ coating antigen. Sera collected from eight MPXV infection-convalescent individuals were tested for the quantification of virus-specific IgG, using sera from normal healthy donors as negative controls. As shown in Figure 1A, all the sera of individuals convalescing from Mpox show high titers of anti-MPXV IgG antibodies compared to the negative sera from the healthy control subjects (Figure 1A).

To validate the results obtained with the homemade ELISA test, we used an antibody-mediated MPXV neutralization assay. Sera of all the individuals convalescent from Mpox neutralize MPXV in vitro with titers ≥ 1:10 (Figure 1B and Table 1). Interestingly, we observed a good correlation (*R*^2^ value = 0.9711) between ELISA MPXV-specific IgG titers and the PRNT50 titers (Figure 1C).

Orthopoxviruses are antigenically and genetically similar, with open reading frames (ORFs) having >90% sequence identity among members. Therefore, MPXV, VV, and Variola Virus (smallpox) share many antigens. For this reason, we were interested in evaluating whether monkeypox-specific IgG induced following infection were able to cross-react with VV proteins. Sera from all the convalescent individuals from an MPXV infection tested positive for anti-VV IgG with titers ranging from 1:40 to 1:6420 (Figure 1D), proving a cross-reactivity between the IgG induced by the MPXV infection and the VV culture supernatant used as coating antigen.

### 3.2. Role of the Residual Immunological Smallpox IgG Memory Response in the Serological Diagnosis of Mpox

A possible limitation of the developed MPXV-specific serologic ELISA test could depend on the fact that, since MPXV is an Orthopoxvirus, the results could be biased due to the VV-specific IgG in subjects who have received the smallpox vaccination. To evaluate whether the VV-specific antibodies, induced by smallpox vaccination more than 45 years before, could still be present in circulation and therefore be able to affect the specificity of the MPXV ELISA, we recruited 41 healthy subjects aged between 46 and 64 years, who received the smallpox vaccination at infancy, and 12 younger healthy individuals (age range: 32–44), who did not receive the smallpox vaccination. We tested these sera in a homemade ELISA test with the vTF7-3 protein lysate from an inactivated VV-culture supernatant as coating antigen. None of the younger volunteers, as well as five smallpox-vaccinated subjects, showed significant VV-specific IgG in their sera. A total of 36 out of the 41 vaccinated individuals had detectable antibodies at different titers, induced by the smallpox vaccination after at least 45 years (ranging from 1:100 to 1:6420, with an average titer of 1:626) (Figure 2A).

Then, we asked whether VV-specific antibodies in the sera from smallpox-vaccinated individuals were endowed with the capacity to bind MPXV in vitro. We could demonstrate that the same 36 smallpox-vaccinated individuals aged > 45 years had anti-VV antibodies cross-reacting with MPXV antigens (Figure 2B). Moreover, sera from vaccinated subjects were able to neutralize MPXV in vitro (Table 1 and Figure 2C). The ability of antibodies in the sera of Mpox-convalescent and VV-vaccinated individuals to recognize proteins in MPXV and VV was evaluated by performing a WB assay. Supernatants from Vero cells that were either infected with MPXV or VV or not infected were subjected to SDS-PAGE, followed by a WB using the sera of Mpox-convalescent patients, smallpox-vaccinated or unvaccinated individuals as control. Figure 2D shows that smallpox-vaccinated individuals aged > 45 years have anti-VV antibodies elicited at infancy that cross-react with the MPXV antigens. Likewise, Mpox-convalescent patients have anti-MPXV antibodies able to bind VV antigens. Interestingly, we could observe a good correlation (*R*^2^ value = 0.6438) between ELISA MPXV-specific IgG titers and PRNT50 titers against MPXV with the sera of smallpox-vaccinated individuals aged > 45 years (Figure 2E).

### 3.3. Possible Role of Polyclonal Non-Specific Activation in the Maintenance of Immunological Memory

Smallpox-vaccinated individuals, with a few exceptions such as laboratory or military personnel, did not receive a natural booster after the primary vaccination more than 45 years ago, since the variola virus was eradicated and no additional vaccine doses were administered due to the end of the global immunization campaign in the 1970s. An analysis of the immune response to smallpox vaccinations is therefore a good tool to explore possible factors affecting immunological memory. To test the role of polyclonal non-specific activation in the maintenance of immunological memory, we compared the anti-VV humoral and cellular immune response in individuals with or without LTBI. As reported in Figure 3, when we divided our study population of vaccinated individuals into those with a positive IGRA and those with a negative IGRA, the mean titer of VV-specific IgG in vaccinated individuals with LTBI was significantly higher than that of donors without LTBI (*p* = 0.0429). We then measured the frequencies of both CD4+ and CD8+ T cells that undergo activation after an in vitro culture with PPD or MVA in PBMC of all the vaccinated individuals. Figure 4 reports a representative flow cytometric analysis showing the activation, measured as a dilution of CFSE and increase in cytokine production, of T lymphocytes three days after stimulation with the indicated antigens. The T lymphocytes of the vaccinated individuals with or without LTBI proliferate and produce cytokines after stimulation with MVA, but only T cells from individuals with LTBI respond to PPD stimulation (Figure 4). Moreover, the frequency and the cytokine production are higher in individuals with LTBI when stimulated with MVA (Figure 5).

## 4. Discussion

More than 94,000 Mpox cases were recorded in the 2022–2023 outbreak (https://www.cdc.gov/poxvirus/mpox/response/2022/index.html, accessed on 10 January 2024). Most cases were in adult males, with a median age (38 years) similar to the age range seen in outbreaks in Africa over the recent years [6]. Laboratory-based diagnosis assumes a critical role for the identification and the subsequent clinical management of suspected Mpox cases. RT-qPCR is the current gold standard technique for the laboratory diagnosis of Mpox, but it has several drawbacks, including a long sample processing time, requirement of technical expertise, reliable access to electricity, and the use of a sophisticated thermocycler for the detection and amplification of the viral genome, all of which may not be easily available in all countries and, in particular, those with a low or very low income [5]. Serological tests, such ELISA, lateral flow assays, plaque reduction neutralization testing (PRNT), immunofluorescence assay, and immunohistochemistry have been developed as potential tools for the diagnosis of Mpox [5]. However, the diagnostic power of these assays in the differentiation of Mpox from acute or past infections due to other Orthopoxviruses is limited. In fact, cross-reactivity between Orthopoxviruses represents one of the most critical limitations of serological-based methods for the diagnosis of Mpox in clinical practice. A potential confounding result of the serological diagnosis of Mpox is also represented by previous vaccinations with VV against smallpox. In Italy, vaccinations with VV, as part of the global immunization campaign aimed at smallpox eradication, ended in the 1970s and vaccinated individuals never received boosters or underwent natural infection in the following 45–60 years after vaccination, with a few possible exceptions. To assess the influence of the residual humoral immune response after a smallpox vaccination at infancy on the serological diagnosis of Mpox, we set up a homemade ELISA and an in vitro viral neutralization assay to test the sera from Mpox-convalescent patients and from individuals vaccinated against smallpox more than 45 years before. Results clearly show that non-vaccinated individuals were negative in both ELISA and the neutralization test, while most individuals vaccinated >45 years ago tested positive in the Mpox serodiagnostic test, still having IgG that was able to bind cross-reactive VV antigens. Sera from Mpox-convalescent individuals had IgG recognizing both MPXV and VV, while smallpox-vaccinated individuals recognized both VV and MPXV antigens in WB, even with a different pattern of bands. As expected, none of the healthy non-smallpox-vaccinated individuals recognized VV or MPXV antigens in the same WBs. These data indicate that tools for serodiagnosis, such as ELISA tests based on MPXV-inactivated supernatants or viral neutralization assays, may be useful for the diagnosis of Mpox in non-endemic areas, such as Europe or the US. In particular, these tests may be useful for retrospective diagnoses, in the absence of viruses in the healed lesions and for seroepidemiologic surveys of viral spread among populations that may include symptomatic or asymptomatic individuals. Further studies will be required to evaluate their value as surrogate markers of protection. However, particular care should be given to ascertain a previous smallpox vaccination or infection that occurred in areas of endemicity of other Orthopoxviruses.

ELISA for IgG specific for VV antigens demonstrated that the majority of individuals vaccinated against smallpox more than 45 years before still have specific antibodies at high titers. Even if the sera of these subjects were able to neutralize MPXV in vitro, it is hard to foresee whether these individuals are still protected from smallpox or Mpox after so many years following the vaccination [8,10]. However, data from a recent publication indicate that a previous smallpox vaccination at a median of 13 years earlier reduced the likelihood of testing positive for the Orthopoxvirus among military personnel [17]. But, irrespective of the potential protection, it is noteworthy that these specific antibodies represent an extraordinary example of long-lasting antigen-specific immunological memory. In fact, we can consider that, with a high grade of confidence, they did not receive any natural boosts after the primary vaccination, since the variola virus is eradicated, and no cases of infections from other Orthopoxviruses are known. Moreover, additional vaccine doses were not administered, due to the end of the global immunization campaign in the 1970s. An analysis of the immune response to smallpox vaccination is therefore a good tool to explore possible factors affecting the immunological memory in the absence of an antigen. According to the polyclonal B cell activation, two types of stimuli can trigger B lymphocyte proliferation and differentiation in plasma-cell-producing antibodies in the absence of an antigen [13]. Microbial products may directly stimulate B cells via the Toll-like receptors, (TLR)4 and TLR9, such as lipopolysaccharide or unmethylated single-stranded DNA motifs, respectively [18,19]. Microbes or other stimuli may also concur indirectly to polyclonal B cell activation through the stimulation of specific T cells that in turn stimulate B cells in a noncognate fashion via CD40 ligand and cytokine production, an interaction that was defined bystander help [13,20]. According to the hypothesis, memory B cells responding to environmental stimuli undergo proliferation and differentiation. In this way, a constant level of plasma cells and serum antibodies could theoretically be maintained throughout a human lifespan. In light of this, the possibility that LTBI could concur to the maintenance of immunological memory against both *Mycobacterium tuberculosis* (Mtb) and other non-related antigens, such as VV antigens, is particularly interesting. It is generally accepted that Mtb in subjects with LTBI survives in a dormant status [21] under the immunological pressure of immunocompetent hosts. However, Mtb occasionally resuscitates as a ‘scout’ bacterium to test the environmental suitability for reactivation [22]. In immunocompromised individuals, the occasional resuscitation of Mtb may represent the trigger for reactivation tuberculosis. On the other hand, Mtb resuscitation from dormancy in immunocompetent individuals causes its recognition by specific T cells that are restimulated, expanded and that presumably kill it or cause its return to a dormant status [23]. Upon Mtb resuscitation in immunocompetent individuals, both types of stimuli that were able to induce polyclonal B cell activation could concur to the proliferation and differentiation of different B cells in plasma cell, even in the absence of the antigen they are specific to. To test this hypothesis, we compared the titers of antibody specific for VV and the frequency and capacity to specifically respond to their antigen of anti-VV-specific T cells in smallpox-vaccinated individuals with or without LTBI. Results showed that individuals with LTBI, identified as such because of a positive Quantiferon^®^ test, displayed mean anti-VV titers that were higher than that of the Quantiferon^®^ negative individuals. As far as specific T cell memory is concerned, only T cells from individuals with LTBI were activated, and secreted cytokines against PPD, as expected. Moreover, VV-specific T cells from individuals with LTBI showed a frequency and an activation capacity higher than those without LTBI. Together, these data indicate that, among subjects vaccinated more than 45 years before, the subpopulation of individuals with LTBI have a more robust immunological memory against VV than the subpopulation of LTBI negative individuals. Even if the sample size of tested individuals is limited, our data support the hypothesis that a “bystander” or “non-specific polyclonal” activation may concur to the maintenance of immunological memory in the absence of an antigen. Even if many factors can lead to differences between the analyzed groups, the observed correlation is corroborated by the notion that no differential post-vaccination exposure to smallpox was possible, and by the observation that the only known relevant difference in the population of smallpox-vaccinated individuals studied is the presence or absence of LTBI. At the recruitment for this, the study donors that were selected included healthy male military personnel that underwent the same vaccination schedule (that did not include a smallpox booster) in their professional life, and their medical history documented an absence of chronic diseases or immunocompromising factors that, being incompatible with the qualification for the military service, would have been noticed.

## 5. Conclusions

Serodiagnosis of Mpox by ELISA and in vitro neutralization may be useful for the identification of infected individuals and for seroprevalence studies. However, to increase the power of these tests, particular care should be given to ascertain a previous smallpox vaccination or infection that had occurred in the areas of endemicity of other Orthopoxviruses. An immune response to VV antigens acquired by smallpox vaccination is, in fact, measurable even after more than 45 years from immunization. Our data, even if in a limited number of subjects, support the theory of “bystander help” or “non-specific polyclonal activation” in the maintenance of immunological memory.

## Figures and Tables

**Figure 1 viruses-16-00620-f001:**
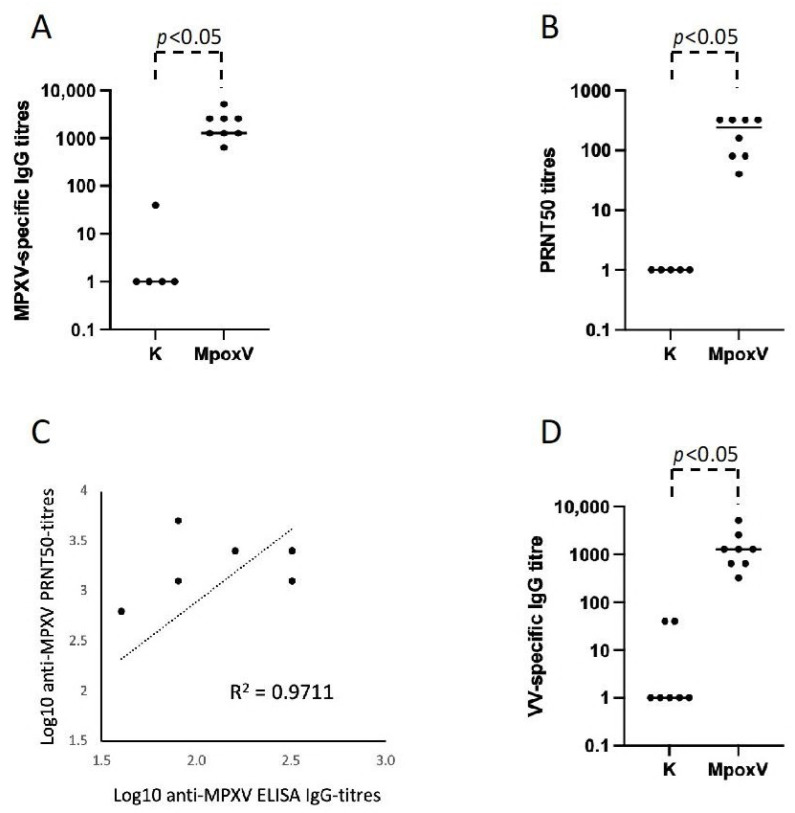
Humoral immune response in Mpox-convalescent individuals. (**A**) Anti MPXV-specific IgG ELISA titers in sera collected from eight Mpox-convalescent individuals (Mpox) compared to healthy negative controls (K). (**B**) Anti-MPXV PRNT50 titers in sera of Mpox-convalescent individuals (Mpox). (**C**) Correlation between ELISA MPXV-specific IgG titers and PRNT50 titers. *R*^2^ indicates the value of the coefficient of determination. (**D**) Anti VV-specific IgG ELISA titers in sera collected from Mpox-convalescent individuals (Mpox) compared to healthy negative controls.

**Figure 2 viruses-16-00620-f002:**
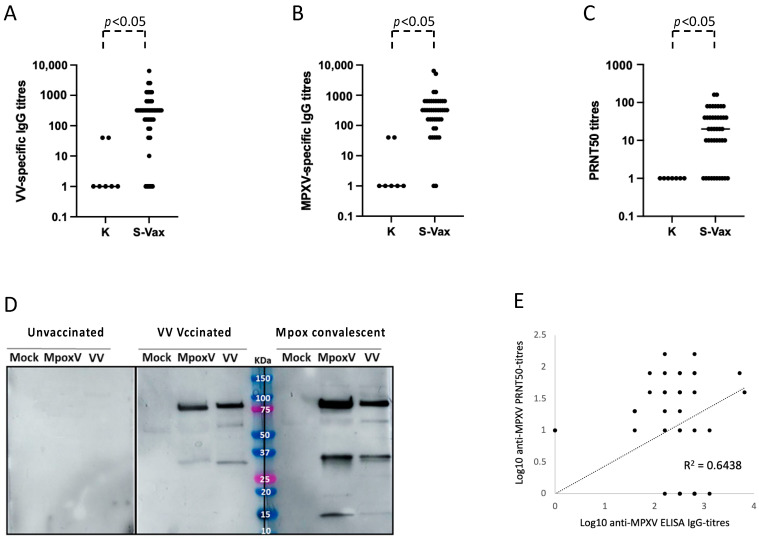
Humoral immune response in smallpox-vaccinated individuals. (**A**) Anti-VV specific IgG ELISA titers in the sera of smallpox-vaccinated subjects (S-Vax) or young unvaccinated controls (K). The line indicates the mean value of VV-specific IgG titers. (**B**) Anti MPXV-specific IgG ELISA titers in the sera collected from 41 smallpox-vaccinated subjects (S-Vax). The line indicates the mean value of MPXV-specific IgG titers. (**C**) VV neutralization assays with the sera of 41 smallpox-vaccinated subjects (S-Vax). The line indicates the mean value of MPXV neutralization titer. (**D**) WB analysis of serum of unvaccinated (left panel), VV-vaccinated (middle panel) and Mpox-convalescent (right panel) individuals in SDS-PAGE supernatants of Vero E6 cells not infected (Mock) or infected with VV or MPXV. A representative experiment of three independent assays is shown. (**E**) Correlation between ELISA MPXV-specific IgG titers and PRNT50 titers against MPXV in the sera of smallpox-vaccinated individuals aged > 45 years. In the figure, the regression line and the *R*^2^ value of coefficient of determination are indicated.

**Figure 3 viruses-16-00620-f003:**
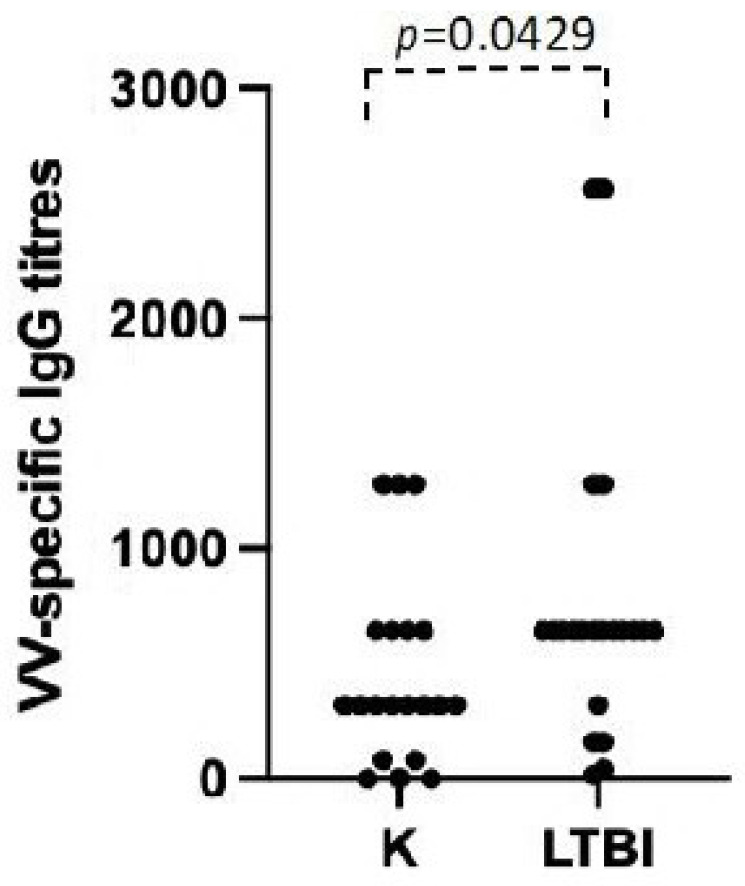
VV-specific ELISA IgG titers in the LTBI subjects and control individuals (K). The *p* value indicates the statistical difference among the titers in the control group and the LTBI subjects.

**Figure 4 viruses-16-00620-f004:**
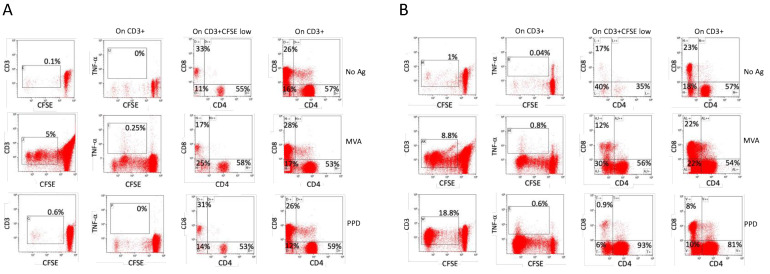
Peripheral CD14^neg^ responder cells from smallpox-vaccinated individuals without (**A**) or with LTBI (**B**), labeled with CFSE, were co-cultured with non-infected or MVA-infected or PPD-pulsed autologous monocytes. After 6 days of co-culture, the T cells were restimulated with autologous monocyte-derived DC infected with MVA or pulsed with PPD or not treated (No Ag). Two days after restimulation, the T cell activation was evaluated by flow cytometric analysis of CFSE dilution and intracellular accumulation of TNF-α. Distributions of CD4^+^ and CD8^+^ cells on all CD3^+^ cells or on CD3^+^CFSE^low^ cells are shown. Numbers indicate the percentages of cells in the corresponding quadrant. One representative experiment of two independent experiments with the same two subjects is shown.

**Figure 5 viruses-16-00620-f005:**
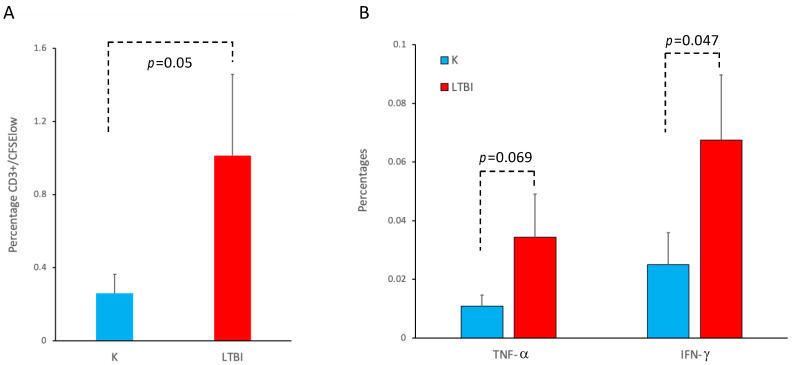
Peripheral CD14^neg^ responder cells from smallpox-vaccinated individuals without (K) or with LTBI (LTBI), labeled with CFSE, were co-cultured with MVA-infected autologous monocytes. After 6 days of co-culture, the T cells were restimulated with autologous monocyte-derived DC infected with MVA. Two days after restimulation, the CD3^+^ T cell activation was evaluated by flow cytometric analysis of CFSE dilution (**A**) and intracellular accumulation of TNF-α or IFN-γ in CD3^+^ CFSE^low^ cells (**B**). The *p* values indicate the statistical differences among smallpox-vaccinated individuals without (K) or with LTBI.

**Table 1 viruses-16-00620-t001:** MPXV PRNT results are shown for smallpox-vaccinated subjects and for Mpox-convalescent patients.

	PRNT50 Positives/Tested	PRNT50 Titers Mean ± SD (Median, Range)	PRNT80 Positives/Tested	PRNT80 Titers Mean ± SD (Median, Range)
Vaccinated (N = 41)	34/41	4.88 ± 1.28 (median 5.32, range 3.32–7.32)	22/41	4.55 ± 0.75 (median 4.32, range 3.32–5.32)
Mpox convalescent patients (N = 8)	8/8	7.32 ± 1.20 (median 7.82, range 5.32–8.32)	8/8	5.20 ± 2.36 (median 3.82, range 3.32–8.32)

## Data Availability

Data supporting the reported results can be acquired at any time from the corresponding author.

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
