# Peer review of "Antibodies Induced by Smallpox Vaccination after at Least 45 Years Cross-React with and In Vitro Neutralize Mpox Virus: A Role for Polyclonal B Cell Activation?"

_viruses, 2024, doi:10.3390/v16040620_

Round 1

Reviewer 1 Report

Comments and Suggestions for Authors

Major remarks:

1.      The claimed association of polyclonal non-specific response is overstated in the title and conclusion. The study requires significantly more information on the subject selection and their demographics to ensure comparison of equal demographics with key confounding factors accounted for.

2.      The number of subjects with MPXV is too small for a meaningful comparison.

Minor remarks:

1.      Antigenic cross-reactivity within the orthopoxviruses is well described, therefore it is not surprising to see the PRNT & ELISA cross-reactivity in this study. Have the authors considered other methodologies, such as cross-absorption ELISA to improve the specificity of the test?

2.      Abstract: (line 33) “Sera from most of the vaccinated individuals showed IgG anti-VV and 33 anti-MPXV at high titers”.  Is it expected to have high titre post vaccination? How does this compared with previous published studies?

3.      Under subheading “Study Population” (line 85): could the authors please provide further demographic information, e.g. age, gender, immunosuppression, whether high-risk group for Mpox has been excluded from the smallpox-vaccinated individuals? What is the distribution of timing of vaccination, since the authors claim “ at least 45 years” in the title. What is the proportion of individuals who fall under this category? Was the healthy control group demographically matched with the vaccinated group?

4.      Only 8 individuals were recruited post MPXV infection. Timing of infection and collection was not stated. This number is too small.

5.      For “VV- or MPXV-specific antibodies” (line 153) under subsection ELISA, it was mentioned that vTF7-3 protein lysate (line 155) was used. However, there was no mention of how VV or MPXV-specific antibodies were determined. Is this by using VV or MPXV-specific antigens or antibodies targeting VV and MPXV specific epitopes? Please elaborate.

6.      Figure 1 C: X-axis (log10 anti-MPXV ELISA IgG-titres): Is this OD value or IgG titres? Have there been any evaluation done to demonstrate correlation of OD and IgG titres within this range?

7.      Figure 1: (A) and (D) should be presented in the same scale on Y-axes

8.      Figure 1(C) and 2(E) showed better R2 in Mpox infected individuals compared with post-vaccination with VV. Please discuss impact of the lack of titre correlation in inferring MPXV-protective immunity  post-vaccination, using MPXV ELISA.

9.      Line 327-329: “Interestingly, we could observe a good 327 correlation (R2 value = 0.6438) between ELISA MPXV-specific IgG titers and PRNT50 titers 328 against MPXV with sera of smallpox vaccinated individuals aged > 45 years (Fig. 2E). “ This is an overclaim.

10.    Line 331: “Possible role of polyclonal non-specific activation in the maintenance of the 331 immunological memory. “

11.    This is an overclaim of the association. I suggest rephrase this as a correlation. There is an observable differences (p=0.04) in control vs LTBI group (Figure 3). However, lots of factors can lead to differences between the groups (e.g. timing of vaccination to collection, medical history, other chronic infections, immunosuppression etc etc), which have not been accounted for. The authors should mention in the discussion the different causes of polyclonal non-specific activation. Similarly, for figure 4, differences in CD14 responder cells do not indicate “causal relationship”.

Author Response

Reviewer 1

Yes

Can be improved

Must be improved

Not applicable

Does the introduction provide sufficient background and include all relevant references?

(x)

( )

( )

( )

Are all the cited references relevant to the research?

(x)

( )

( )

( )

Is the research design appropriate?

( )

( )

(x)

( )

Are the methods adequately described?

( )

(x)

( )

( )

Are the results clearly presented?

( )

(x)

( )

( )

Are the conclusions supported by the results?

( )

( )

(x)

( )

Comments and Suggestions for Authors

Major remarks:

  1. The claimed association of polyclonal non-specific response is overstated in the title and conclusion. The study requires significantly more information on the subject selection and their demographics to ensure comparison of equal demographics with key confounding factors accounted for.

The results of our work are suggestive for a role of the polyclonal non-specific response in the maintenance of immunological memory in the absence of antigen. The title stresses this suggestion with a question mark: “…a role for polyclonal B cell activation?”. Moreover, in the conclusion we wrote “..our data support the hypothesis that a “bystander” or “non-specific polyclonal” activation may concur to the maintenance of immunological memory in the absence of antigen”. We do not believe that these sentences overstate an association, rather suggest a possible correlation that supports previous studies from other authors. In any case a more detailed description of subject’s selection and their demographics has been included in the study population chapter.

  1. The number of subjects with MPXV is too small for a meaningful comparison.

We are aware that the number of subjects convalescent from Mpox is limited, but we think it may be adequate, when used in comparison to sera of non-infected individuals, for the scope of validating ELISA and PRNT of antibodies elicited by smallpox vaccination more than 45 years before the test.

Minor remarks:

  1. Antigenic cross-reactivity within the orthopoxviruses is well described, therefore it is not surprising to see the PRNT & ELISA cross-reactivity in this study. Have the authors considered other methodologies, such as cross-absorption ELISA to improve the specificity of the test?

We considered the analysis in Western Blot as an adequate tool to confirm the specificity of our ELISA and PRNT for the two viruses.

  1. Abstract: (line 33) “Sera from most of the vaccinated individuals showed IgG anti-VV and 33 anti-MPXV at high titers”.  Is it expected to have high titre post vaccination? How does this compared with previous published studies?

We have included a citation of the study: Matusali, G.,et al. "Evaluation of Cross-Immunity to the Mpox Virus Due to Historic Smallpox Vaccination." Vaccines (Basel) 11, no. 10 (2023) that reports data in line with our results.

  1. Under subheading “Study Population” (line 85): could the authors please provide further demographic information, e.g. age, gender, immunosuppression, whether high-risk group for Mpox has been excluded from the smallpox-vaccinated individuals? What is the distribution of timing of vaccination, since the authors claim “ at least 45 years” in the title. What is the proportion of individuals who fall under this category? Was the healthy control group demographically matched with the vaccinated group?

Additional information has been included in the text to better clarify the studied population.

  1. Only 8 individuals were recruited post MPXV infection. Timing of infection and collection was not stated. This number is too small.

The number of sera from Mpox convalescent patients is small but we think sufficient when used in comparison to sera of non-infected individuals for the scope of validating ELISA and PRNT. Timing of infection and collection is now reported.

  1. For “VV- or MPXV-specific antibodies” (line 153) under subsection ELISA, it was mentioned that vTF7-3 protein lysate (line 155) was used. However, there was no mention of how VV or MPXV-specific antibodies were determined. Is this by using VV or MPXV-specific antigens or antibodies targeting VV and MPXV specific epitopes? Please elaborate.

We apologize for an error in the description of the ELISA method that may have generated the correct criticism of the reviewer. We have now changed as follows: ”Briefly, 96-well polystyrene plates (Greiner Bio-One, Rainbach, Austria) were coated with 50 μL/well of vTF7-3 or hMpxV/Italy protein lysate at a concentration of 5 μg/mL in carbonate bicarbonate buffer and incubated at +4 °C overnight.” Then we used sera from smallpox vaccinated or Mpox convalescent patients, respectively, in comparison to non-smallpox vaccinated and MPXV negative individuals, to set up an ELISA able to discriminate the population. Moreover, a WB assay confirmed the specificity. 

  1. Figure 1 C: X-axis (log10 anti-MPXV ELISA IgG-titres): Is this OD value or IgG titres? Have there been any evaluation done to demonstrate correlation of OD and IgG titres within this range?

The figure reports the correlation between anti-MPXV ELISA IgG-titres and anti-MPXV neutralization titres. We did not perform any evaluation to correlate OD and IgG titres. Titres were calculated based on OD values in serial dilution, as now specified in M&M: “Absorbance (405 nm) was measured, and the results were considered positive if the optical density (OD) was three times greater than the negative control. For each serum, the titre corresponds to the last dilution giving a positive result.”

  1. Figure 1: (A) and (D) should be presented in the same scale on Y-axe

This figure now reports data within the same scale on Y axe.

  1. Figure 1(C) and 2(E) showed better R2 in Mpox infected individuals compared with post-vaccination with VV. Please discuss impact of the lack of titre correlation in inferring MPXV-protective immunity post-vaccination, using MPXV ELISA.

There was a stronger correlation between MPXV ELISA titres of sera from Mpox convalescent individuals and MPXV neutralization assay than between MPXV ELISA titres of sera from VV vaccinated individuals and the same MPXV PRNT-50. Since MPXV PRNT-50 titres were higher than MPXV ELISA titres, this result suggests that the mean concentration of neutralizing antibodies cross-reacting with MPXV antigens that is required to inhibit the infection is higher than the concentration of antibodies detected by the MPXV ELISA in sera from VV vaccinated individuals. As noted in the Discussion section, MPXV ELISA may be useful for retrospective diagnoses, in the absence of viruses in the healed lesions and for seroepidemiologic surveys of viral spread among populations that may include symptomatic or asymptomatic individuals, but further studies will be required to evaluate their value as surrogate markers of protection.

  1. Line 327-329: “Interestingly, we could observe a good 327 correlation (R2 value = 0.6438) between ELISA MPXV-specific IgG titers and PRNT50 titers 328 against MPXV with sera of smallpox vaccinated individuals aged > 45 years (Fig. 2E). “ This is an overclaim.

We agree with the reviewer and the text has been changed in :” We could also observe a correlation (R2 value = 0.6438) between ELISA MPXV-specific IgG titers and PRNT50 titers against MPXV with sera of smallpox vaccinated individuals aged > 45 years (Fig. 2E)

  1. Line 331: “Possible role of polyclonal non-specific activation in the maintenance of the immunological memory“

This is an overclaim of the association. I suggest rephrase this as a correlation. There is an observable differences (p=0.04) in control vs LTBI group (Figure 3). However, lots of factors can lead to differences between the groups (e.g. timing of vaccination to collection, medical history, other chronic infections, immunosuppression etc etc), which have not been accounted for. The authors should mention in the discussion the different causes of polyclonal non-specific activation. Similarly, for figure 4, differences in CD14 responder cells do not indicate “causal relationship”.

We thank the reviewer for having motivated us to better indicate the reasons why we believe that our data support the hypothesis of a role for the polyclonal non-specific activation in the maintenance of the immunological memory and we expanded our discussion as follows: “Even if many factors can lead to differences between the analyzed groups, the observed correlation is corroborated by the notion that no differential post-vaccination exposure to smallpox was possible, and by the observation that the only known relevant difference in the population of smallpox vaccinated individuals studied is the presence or absence of LTBI. At the recruitment for this study donors were selected as healthy male military personnel that underwent the same vaccination schedule (that does not include smallpox booster) in their professional life and their medical history documented absence of chronic diseases or immunocompromising factors that, being incompatible with the qualification for the military service, would have been noticed”. We agree with Reviewer-1 when he/she writes that: “differences in CD14  responder cells do not indicate “causal relationship”. Indeed, when we expanded the representative experiment reported in figure 4 and reported in figure 5 the results in all the donors, we noted differences in the response to vaccinia virus by CD3 positive T cells (CD14 neg) in terms of activation and cytokine secretion between individuals with ir without LTBI that are suggestive for an increased memory response in those with LTBI. We did not claim any causal relationship, but we described the results indicating that: “the subpopulation of individuals with LTBI have a more robust immunological memory against VV than the subpopulation of LTBI negative individuals”.

Reviewer 2 Report

Comments and Suggestions for Authors

The study presented was well ritten and has its own scientific soundness. Even if not original, this work can be published because it offers readers a simple and clear interpretative key, as well as useful for epidemiological purposes and for understanding the immune response to vaccinations. Finally, it can provide motivation for the general population to get vaccinated for vaccine-preventable diseases.

Author Response

Reviewer 2

Yes

Can be improved

Must be improved

Not applicable

Does the introduction provide sufficient background and include all relevant references?

(x)

( )

( )

( )

Are all the cited references relevant to the research?

(x)

( )

( )

( )

Is the research design appropriate?

(x)

( )

( )

( )

Are the methods adequately described?

(x)

( )

( )

( )

Are the results clearly presented?

(x)

( )

( )

( )

Are the conclusions supported by the results?

(x)

( )

( )

( )

Comments and Suggestions for Authors

The study presented was well written and has its own scientific soundness. Even if not original, this work can be published because it offers readers a simple and clear interpretative key, as well as useful for epidemiological purposes and for understanding the immune response to vaccinations. Finally, it can provide motivation for the general population to get vaccinated for vaccine-preventable diseases.

We thank the reviewer for his/her comments.

Reviewer 3 Report

Comments and Suggestions for Authors

The discontinuation of smallpox following the eradication of smallpox has resulted in an increased risk of Mpox transmission to vaccine naive individuals. In particular, it is extremely important to verify the degree of immunity to Mpox among subjects who received smallpox in the past, and to understand the relationship with the immunogenicity of the vaccine strains used in the past is essential for future diagnosis and prevention of Mpox.

The authors are conducting scientific validation of these points, but the following points need to be clarified.

L85 What is used as an existing smallpox vaccine can significantly alter the response to Mpox. The authors should identify which vaccine strains the subjects have received in the past. Usually, the vaccine strain used can be ascertained from the date of birth.

L92: What is the reason for the use of the term "prior to overseas deployment?" It is similar to the JAMA article, and if it was referenced, it should be noted in the references. Since this study was not conducted using a smallpox vaccine, there is no need to mention it here.

https://jamanetwork.com/journals/jama/fullarticle/183542

Author Response

Reviewer 3

Yes

Can be improved

Must be improved

Not applicable

Does the introduction provide sufficient background and include all relevant references?

( )

(x)

( )

( )

Are all the cited references relevant to the research?

( )

(x)

( )

( )

Is the research design appropriate?

( )

( )

(x)

( )

Are the methods adequately described?

( )

( )

(x)

( )

Are the results clearly presented?

( )

(x)

( )

( )

Are the conclusions supported by the results?

( )

(x)

( )

( )

Comments and Suggestions for Authors

The discontinuation of smallpox following the eradication of smallpox has resulted in an increased risk of Mpox transmission to vaccine naive individuals. In particular, it is extremely important to verify the degree of immunity to Mpox among subjects who received smallpox in the past, and to understand the relationship with the immunogenicity of the vaccine strains used in the past is essential for future diagnosis and prevention of Mpox.

The authors are conducting scientific validation of these points, but the following points need to be clarified.

 L85 What is used as an existing smallpox vaccine can significantly alter the response to Mpox. The authors should identify which vaccine strains the subjects have received in the past. Usually, the vaccine strain used can be ascertained from the date of birth.  

We have specified that “…all subjects were presumptively vaccinated with the vaccinia virus Lister strain used in Italy in the 1960s and early 1970s produced by the Company Sclavo (Siena, Italy)

 L92: What is the reason for the use of the term "prior to overseas deployment?" It is similar to the JAMA article, and if it was referenced, it should be noted in the references. Since this study was not conducted using a smallpox vaccine, there is no need to mention it here. https://jamanetwork.com/journals/jama/fullarticle/183542

We have explained what was meant for prior to overseas deployment: “All the volunteers were recruited from military personnel of Italian Air Force, and blood samples were collected during check-up carried out from January and September 2018, before their deployment in specific missions abroad, including International Peacekeeping Operation activities.”

Reviewer 4 Report

Comments and Suggestions for Authors

This manuscript presents important data that demonstrate antibodies against vaccinia vaccination more than 40 years in the past can provide neutralization against monkeypox (mpox).  The volunteers were recruited from members of the Italian Air Force.  The authors also tested for latent TB infection in all volunteers.  The volunteers were divided into 4 groups, with a total of 61 subjects.  Antibody titers from the 41 vaccinated patients confirmed long-lasting memory of specific IgG antibodies that reacted with vaccinia virus and cross-reacted with mpox.  One new reference is recommended, because it is strongly supportive.  

1.  Methods.  Lines 86-88.  State the precise dates (month/year) when the first blood sample and the last blood sample were collected from the donors in the Air Force.

2.  Results.  Lines 332-336.  In the United States, first-responders, including many military personnel, were offered a booster vaccinia vaccination after the 9/11 terrorist attacks in the USA in 2001.  Please state at this location in text or in the Methods that none of the 66 volunteers in the study had received a vaccinia vaccination in the 21st century because of possible involvement in anti-terrorist military activities in Italy.

3.  Results, lines 285-299.  Figure 2.  These data are extremely important.  Instead of using the word majority in line 296, state the actual number of volunteers with antibody against vaccinia (Fig. 2A).  Likewise, in lines 316-318, state the actual number of volunteers that have antibody reacting with mpox (Fig. 2B).

4.  Discussion.  Add a new reference in section 5.  This reference from the Veterans Hospitals in the United States found that Dryvax vaccination 13 years before exposure to mpox was protective.  This short article strongly supports the results of the Italian study.  See article by B. Titanji et al, Effectiveness of smallpox vaccination to prevent mpox in military personal, New England J. Medicine 389:1147. 2023  (PMID:37733313).  Briefly mention this supportive paper in Section 5.   

Author Response

Reviewer 4

Yes

Can be improved

Must be improved

Not applicable

Does the introduction provide sufficient background and include all relevant references?

(x)

( )

( )

( )

Are all the cited references relevant to the research?

(x)

( )

( )

( )

Is the research design appropriate?

(x)

( )

( )

( )

Are the methods adequately described?

(x)

( )

( )

( )

Are the results clearly presented?

(x)

( )

( )

( )

Are the conclusions supported by the results?

(x)

( )

( )

( )

Comments and Suggestions for Authors

This manuscript presents important data that demonstrate antibodies against vaccinia vaccination more than 40 years in the past can provide neutralization against monkeypox (mpox).  The volunteers were recruited from members of the Italian Air Force.  The authors also tested for latent TB infection in all volunteers.  The volunteers were divided into 4 groups, with a total of 61 subjects.  Antibody titers from the 41 vaccinated patients confirmed long-lasting memory of specific IgG antibodies that reacted with vaccinia virus and cross-reacted with mpox.  One new reference is recommended, because it is strongly supportive.  

  1. Methods.  Lines 86-88.  State the precise dates (month/year) when the first blood sample and the last blood sample were collected from the donors in the Air Force.

The time frame for blood sample collection has now been included in the text:” blood samples were collected during check-up carried out from January and September 2018”.

  1. Results.  Lines 332-336.  In the United States, first-responders, including many military personnel, were offered a booster vaccinia vaccination after the 9/11 terrorist attacks in the USA in 2001.  Please state at this location in text or in the Methods that none of the 66 volunteers in the study had received a vaccinia vaccination in the 21stcentury because of possible involvement in anti-terrorist military activities in Italy.

We have now better specified that none of the enrolled volunteers received booster(s) of smallpox vaccine.

  1. Results, lines 285-299.  Figure 2.  These data are extremely important.  Instead of using the word majority in line 296, state the actual number of volunteers with antibody against vaccinia (Fig. 2A).  Likewise, in lines 316-318, state the actual number of volunteers that have antibody reacting with mpox (Fig. 2B).

The actual number of volunteers with antibody against vaccinia virus has now been included in the Results section.

  1. Discussion.  Add a new reference in section 5.  This reference from the Veterans Hospitals in the United States found that Dryvax vaccination 13 years before exposure to mpox was protective.  This short article strongly supports the results of the Italian study.  See article by B. Titanji et al, Effectiveness of smallpox vaccination to prevent mpox in military personal, New England J. Medicine 389:1147. 2023  (PMID:37733313).  Briefly mention this supportive paper in Section 5.   

We thank the reviewer for the suggestion, and the reference has now been included in the Discussion.

Round 2

Reviewer 1 Report

Comments and Suggestions for Authors

No further comments.